

# Mechanisms and physiological relevance of acid-base exchange in functional units of the kidney

Elena Gantsova[1,2], Oxana Serova[3], Polina Vishnyakova[2,4], Igor Deyev[3], Andrey Elchaninov[1,2] and Timur Fatkhudinov[1,2]

[1] Avtsyn Research Institute of Human Morphology of Federal State Budgetary Scientific Institution "Petrovsky National Research Centre of Surgery", Moscow, Russia
[2] Research Institute of Molecular and Cellular Medicine, Peoples' Friendship University of Russia (RUDN University), Moscow, Russia
[3] Shemyakin–Ovchinnikov Institute of Bioorganic Chemistry, Russian Academy of Sciences, Moscow, Russian Federation
[4] National Medical Research Center for Obstetrics, Gynecology and Perinatology Named after Academician V.I. Kulakov of Ministry of Healthcare of Russian Federation, Moscow, Russian Federation

## ABSTRACT

This review discusses the importance of homeostasis with a particular emphasis on the acid-base (AB) balance, a crucial aspect of pH regulation in living systems. Two primary organ systems correct deviations from the standard pH balance: the respiratory system *via* gas exchange and the kidneys *via* proton/bicarbonate secretion and reabsorption. Focusing on kidney functions, we describe the complexity of renal architecture and its challenges for experimental research. We address specific roles of different nephron segments (the proximal convoluted tubule, the loop of Henle and the distal convoluted tubule) in pH homeostasis, while explaining the physiological significance of ion exchange processes maintained by the kidneys, particularly the role of bicarbonate ions ($HCO_3^-$) as an essential buffer system of the body. The review will be of interest to researchers in the fields of physiology, biochemistry and molecular biology, which builds a strong foundation and critically evaluates existing studies. Our review helps identify the gaps of knowledge by thoroughly understanding the existing literature related to kidney acid-base homeostasis.

## INTRODUCTION

Homeostasis is defined as the constancy of internal medium essential for living systems. The 'constancy' refers to narrow ranges acceptable for particular physical and chemical parameters of the body, the precision of which depends on distinct, finely coordinated functionalities. Each facet of homeostasis comprises a hierarchical network of sensory, executive and effector capacities to ensure rapid adaptation to fluctuating environments.

The acid-base (AB) balance is a critical facet of homeostasis, as the binding properties of biomolecules (key to every biological process) strongly depend on acidity of the aqueous

Corresponding author
Elena Gantsova,
gantsova@MAIL.RU

media of cells and tissues, determined by production and excretion rates of acids and bases, their preservation efficiency and physiological adequacy.

The human body has an evolutionary refined capability of pH adjustment, effectuated by the lungs *via* gas exchange and by the kidneys *via* fine tuning of proton/bicarbonate secretion and reabsorption. In this article we omit the respiratory leverage and focus on the compensatory mechanisms provided by the kidneys. The physiological levels of free $H^+$ ions are low compared with $Na^+$ or $K^+$; accordingly, their role in electrochemical gradients is minimal. The crucial physiological relevance of AB homeostasis is due to its decisive influence on the mobility of ionized compounds and the conformation of proteins including enzymes, receptors, transporters, channels, *etc.* (*Hamm, Nakhoul & Hering-Smith, 2015*). Cell damage in severe acidosis/alkalosis can be triggered by immediate shifts in distributions of potassium, sodium and calcium ions.

The kidneys may seem to play a supporting role in AB homeostasis, as they provide slower correction rates for metabolic imbalances compared with the respiratory system (taking hours-to-days *vs* minutes-to-hours, respectively). However, patients with chronic renal failure develop constant progressive metabolic acidosis (*Wesson, 2021*; *Imenez Silva & Mohebbi, 2022*). An imbalanced AB status has severe neurological consequences; the symptoms include confusion and fatigue for acidosis (*Kraut & Madias, 2012*) and muscle twitching, nausea and overexcitability for alkalosis (*Serova et al., 2020*).

Several physiological ion exchange processes continuously maintained in the kidneys are instrumental for AB homeostasis. The most extensive buffer system in the body is provided by bicarbonate ions ($HCO_3^-$). This system is replenished by carbon dioxide and water reacting to form carbonic acid that dissociates into H+ and bicarbonate. Spontaneous non-catalytic reaction between carbon dioxide and water is slow and physiologically irrelevant; when catalyzed by carbonic anhydrases, the reaction becomes instrumental: fast and also regulatable. Under physiological conditions, the conversion is limited by $CO_2$ availability adjusted *via* breathing modes; the kidneys contribute to AB balance by tuning the rates of $H^+/HCO_3^-$ retention and excretion (*Clayton-smith, 2021*).

The kidneys regulate ion, AB and water balances by means of filtration, reabsorption, secretion and excretion in renal tubules. Each human kidney comprises about $2 \cdot 10^6$ nephrons (Fig. 1) engaged in blood filtration followed by regulated modification of the primary urine by selective reabsorption and secretion. The proximal convoluted tubule reabsorbs the major part of filtered ions, water and nutrients, while the distal tubule and collecting ducts perform selective reabsorption and excretion (controlled by hormones) to modify the final composition of urine depending on homeostatic demands.

Experimental studies on renal functionalities involve 2D and 3D tissue culture methods and *in vivo* studies in rodents. The tissue culture models have limited utility in reproducing the native topology of the kidney with epithelial barriers and gradients; however, pluripotent stem cells can be *in vitro* differentiated into tubular 'organoids' that express epithelial cell markers of renal tubules. The *in vitro* 'organoids' were successfully used to model a number of renal conditions including nephrotoxin-induced injuries and complex diseases such as chronic renal failure involving multiple cell types. The models can be further upgraded through the use of genome editing (*e.g.*, CRISPR-Cas9-mediated). These

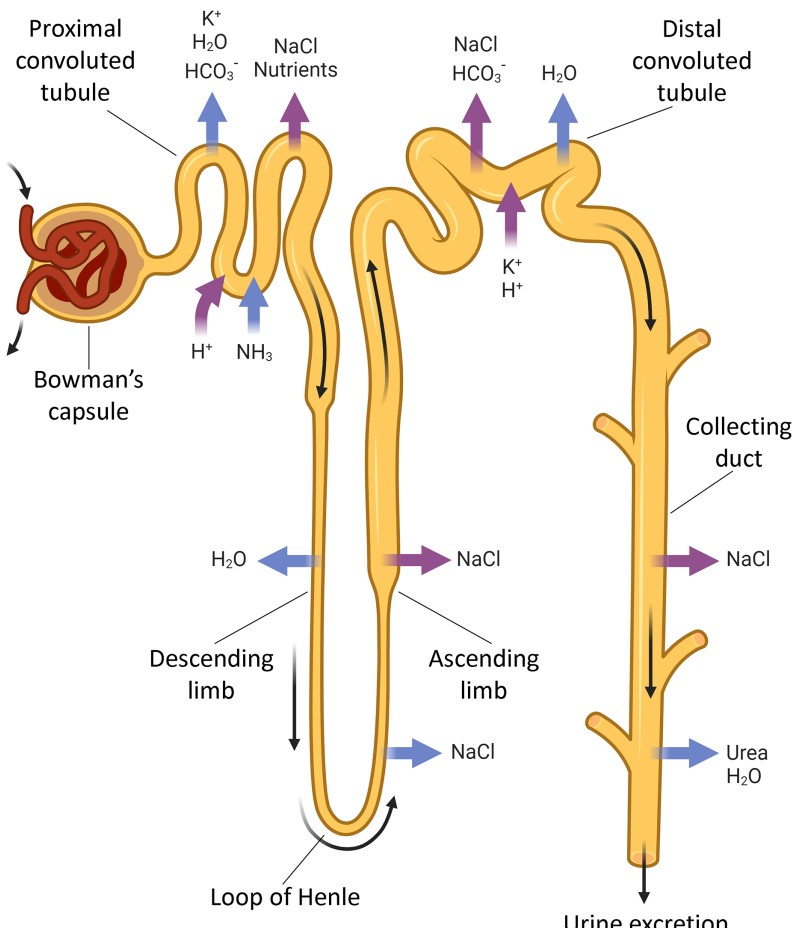

**Figure 1 The nephron parts that are connected with acid-base exchange.** Acid-base exchange, or renal acid-base regulation, is a process in which the kidneys excrete acid, such as hydrogen ions (H⁺) and waste in urine, and absorb bases, such as bicarbonate ions (HCO₃⁻), when needed. The main parts of the nephron involved in acid-base exchange include the proximal tubules, the distal tubules, and the collecting ducts. These segments contribute to the acid-base balance by coordinatedly regulated reabsorption and excretion of bicarbonate and hydrogen ions, as well as other electrolytes that impact the acid-base status. BioRender (biorender.com) was used for the creation of the figure.

findings advance the stem cell-based options as a better alternative to animal models in drug development (*Kishi et al., 2021*).

However, the use of animal models is justified by the common nature of many diseases in humans and animals, including microbial infections, allergies, diabetes, renal failure, hypertension, cancers, myopathies, epilepsies, *etc*. The mechanistic similarity can be illustrated by the fact that about 90% of drugs used in animals are identical or highly similar to those used for corresponding conditions in humans (*Barré-Sinoussi & Montagutelli, 2015*). Still, the interspecies differences are too profound for straightforward translation of clinical experimental data, especially in cases of subtle effects observed on limited samples, even with genetically uniform breeds and strains of laboratory animals (*Miyoshi et al., 2020*). Established mouse models for studying renal pathophysiology cover

a range of conditions including diabetic nephropathy (*Thibodeau et al., 2014*), chronic kidney disease (*Wicks et al., 2016*) and polycystic kidney disease (*Starremans et al., 2008*).

This review is the first in many years to examine the contribution to AB homeostasis for each portion of the nephron specifically. Apart from the main ions responsible for AB balance, we consider renal exchange for some accessory ions including phosphate, ammonium and oxalate. We additionally focus on metabolic disorders related to AB exchange in the kidney and addressed in both mechanistic and clinical aspects of the kidney involvement in AB homeostasis.

## SURVEY METHODOLOGY

Several steps were taken to ensure comprehensive and unbiased coverage of the literature. First, a thorough search was conducted across various academic databases, including PubMed Central, Google Scholar to identify relevant studies, guidelines, and reports. Multiple search terms, such as "kidney physiology", "acid-base homeostasis" and "kidney pH regulation" were used to capture a wide range of sources.

Next, the selected articles and papers were critically reviewed to assess their relevance and quality. To avoid bias, studies from different disciplines and countries were included, considering both theoretical and applied perspectives. Additionally, efforts were made to ensure the inclusion of both classic and contemporary sources, thereby providing a comprehensive overview of the literature.

To further enhance comprehensiveness, a snowballing technique was employed, where the reference lists of the identified articles were examined for additional relevant publications. This helped in discovering additional sources that may not have been captured in the initial search.

### Mechanistic topography of acid-base exchange in the kidney

Systemic AB imbalances manifest as changes in blood plasma pH which only loosely correspond to local shifts in acidity inside cells or at their surface. The normal acidity of intracellular and interstitial compartments may differ from physiological pH of the blood (7.2–7.4) (*Serova et al., 2020*); the differences reflect high rates of transmembrane ion transport in certain cell types or sharp boundaries with harsh external media characteristic of the gastrointestinal or urinary tract epithelial linings.

Extracellular fluids constitute about one-third of the liquid content of the body; the other two-thirds are contained in cells (*Chen, Higgins & Zhang, 2017*). The distribution of extracellular fluids in the body is non-uniform: apart from blood plasma, cerebrospinal fluid and pleuroperitoneal fluid, these include interstitial fluids that closely interact with diverse cell niches in renal tubules, gland acini and ductal systems, lymphoid patches, *etc.* (*Seifter & Chang, 2017*). Metabolic alterations to interstitial fluids can be deteriorating.

The acid and base contents of the body are regulated separately. The baseline plasma levels of inorganic ions are $HCO_3^-$ 27 mmol/L, $Cl^-$ 103 mmol/L, $SO_4^{2-}$ 0.5 mmol/L and $NH_4^+$ 40 mmol/L (*Nezafati, Moztarzadeh & Hesaraki, 2012*; *Azagra et al., 2022*). The acid balance involves three major components: (1) production of $H^+$ in the liver through oxidation of digested nutrient proteins; (2) titration of these protons with bicarbonate ions

$(HCO_3^-)$; and (3) production of new bicarbonate ions accompanied by excretion of ammonium $(NH_4^+)$ with the urine. Phosphoric and sulfuric acids are the primary inorganic acids obtained from foods and processed in the kidneys. The protons of sulfuric acid, $H_2SO_4$, formed by oxidation of sulfur-containing amino acids in the liver, can only be eliminated through ammonium $(NH_4^+)$ excretion, since $SO_4^{2-}$ ions have low affinity for $H^+$. This concept of acid balance is supported by similar daily excretion rates for $NH_4^+$ and $SO_4^{2-}$ (in milliequivalents). Thus, the kidneys should respond to chronic acid overloads by increased excretion of $NH_4^+$ into the lumina of proximal convoluted tubules or medullary collecting tubules. The so-called 'ammonia shunt' from the loop of Henle to medullary collecting tubules (*Halperin, 1989*; *Nagami, 1989*) may negligibly contribute to $NH_3$ excretion, as its main physiological role is to prevent a decrease in pH of the urine through distal secretion of protons thereby minimizing the risks of uric acid precipitation.

The fruit- and vegetable-rich diets increase the bicarbonate/base loads; the excess is subject to clearance for the sake of proper base balance. This physiological task is apparently solved through increased bicarbonate excretion by the kidneys (*Cheema-Dhadli, Lin & Halperin, 2002*). With pH close to 6.0 most of the 24-h cycle, the urine contains very low levels of $HCO_3^-$. Importantly, there is no renal threshold or tubular maximum for $HCO_3^-$ reabsorption. As plasma levels of $HCO_3^-$ increase each time HCl is released into the stomach, a saturation limit for $HCO_3^-$ reabsorption initially hypothesized by *Pitts & Lotspeich (1946)* would lead to bicarbonaturia. However, the release of HCl promotes no increase in extracellular fluid volume, no loss of $Cl^-$ and no increase in $HCO_3^-$ levels. By contrast, in experimental settings, the extracellular fluid volume increases. Administration of $NaHCO_3$ studied by *Kurtzman (1970)* led to a decrease in the tubular maximum of $HCO_3^-$ reabsorption during the alkaline flush in the proximal convoluted tubules and an apparent renal threshold for $HCO_3^-$.

The dietary alkaline load is initially converted to bicarbonate in the liver. The process is accompanied by production of organic acids titrated by $HCO_3^-$; accordingly, the base balance is maintained through excretion of organic anions combined to $K^+$. This mode of base balance neutralizes the less specific endogenous acid production simultaneously compensating for the extensive intake of alkali with foods. To fully account for the role of the kidneys in AB balance, it is important to consider the excretion of the daily dietary alkaline load in the form of organic anions that can be converted to $HCO_3^-$ within the body.

Suppression of the alkaline loads should avoid the risks of $CaHPO_4$ production and kidney stone formation. The mechanism involving organic anions, notably citrate, not only affords the restoration of base balance while maintaining the urine pH around 6.0, but also minimizes the risks of stone formation through the chelating action of citrate on ionized calcium. The $H^+/K^+$-ATPase important in proton excretion also plays a key role in $K^+$ metabolism of the medulla. Kidney ATPases will be considered in upcoming sections (*Halperin, Cheema Dhadli & Kamel, 2006*).

The functional excretory system includes several anatomical organ systems: the respiratory, digestive, and urinary systems. Each of these systems is involved in maintaining acid-base balance in the body to a greater or lesser extent. In this review, we focus on the role of the urinary system in the regulation of AB balance.

The structural and functional unit of the kidney parenchyma, as already mentioned, is the nephron, within which the main processes of urine formation are carried out. In addition, the nephron is functionally and anatomically adjoined to the nephron by the collecting ducts, which also participate in the regulation of the final composition of secondary urine. In addition, functionally and anatomically, the collecting ducts are adjacent to the nephron and are also involved in the regulation of the final composition of secondary urine. Further, the authors will consider how each of the nephron parts affects the acid-base balance.

### Proximal convoluted tubule

The proximal convoluted tubule (PCT) is engaged in reabsorption and recycling of various ions and solutes. About two-thirds of filtered water, NaCl, $Ca^{2+}$, as well as all glucose, phosphates and amino acids, are reabsorbed by extensive apparatus of specialized transporters acting at the brush-bordered apical and the basolateral surfaces of the epithelial cells. Different PCT segments are coordinated by the electrochemical gradient created by basolateral $Na^+$-$K^+$-ATPase (*Van Der Wijst et al., 2019*). These transport processes require high amounts of energy in the form of ATP generated by abundant elongated mitochondria of the epithelial cells (*Kriz & Kaissling, 2007*). In addition, PCTs reabsorb substantial amounts of albumins and low-molecular-weight plasma proteins, *e.g.*, hormones, enzymes, lipoproteins, vitamin carriers, passing through the glomerular basement membrane into the filtrate. Normally, human urine is almost totally cleared of plasma proteins by the clathrin-dependent endocytosis-mediated reabsorption in PCT (*Eshbach & Weisz, 2017*).

The glomerular filtrate that contacts the apical surfaces of PCT cells is initially similar in composition to the interstitial fluid that contacts the basolateral surfaces. With the total filtrate volume of about 180 liters per day, almost the entire quantity of bicarbonate ion must be reabsorbed from the filtrate. The $HCO_3^-$ reabsorption is tightly coupled to $Na^+$ and water reabsorption. The transport of $HCO_3^-$, about 80% of which occurs in PCT, is provided by two pathways. The major indirect $CO_2$ pathway has been conventionally considered as the unique route of $HCO_3^-$ reabsorption, with $H^+$ is secreted into the lumen *via* the apical $Na^+/H^+$ exchanger NHE3 and the vacuolar-type $H^+$-ATPase. Secreted $H^+$ is combined with $HCO_3^-$ and then converted to $CO_2$ and $H_2O$ by carbonic anhydrase IV; $CO_2$ then enters PCT cells by diffusion to recreate $HCO_3^-$ and $H^+$; the reaction is catalyzed by carbonic anhydrase II. The other pathway is direct $HCO_3^-$ transport into PCT cells by the apically expressed $Na^+/HCO_3^-$ cotransporter NBCn2 (*Guo et al., 2017*); $HCO_3^-$ is further transferred to the blood *via* the basolateral $Na^+/HCO_3^-$ cotransporter NBCe1-A (*Rajkumar & Pluznick, 2018*). The NHE3 sodium/proton exchanger and $H^+$-ATPase located at the apical membranes of PCT cells acidify the lumen and alkalize the cells thereby ensuring about 80% of bicarbonate reabsorption. Transport of bicarbonate ions *via* the basolateral membrane to the interstitium by NBC, a $1Na^+/3HCO_3^-$ transporter, maintains the pH gradient (*Boron, 2006*; *Romero et al., 2013*). The PCT cells also produce new bicarbonate to neutralize the metabolically released mineral acids. The remaining $HCO_3^-$ ions are reabsorbed in the ascending loop of Henle (10%) and distal tubules (10%). Apart from

carbonic anhydrase and NHE3 sodium/proton exchanger, the process involves the proton-pumping ATPase V. From the interstitial fluid $HCO_3^-$ is pumped to the blood mostly by NBCe1-A $Na^+/HCO_3^-$ electrogenic co-transporter, renal splice variant. The protons released by PCT cells into the tubular lumen react with $HCO_3^-$ in the filtrate to form $CO_2$ and $H_2O$; the reaction is catalyzed by carbonic anhydrase IV expressed at the apical membrane. Water molecules cross the apical membrane exclusively *via* aquaporin (AQP1) channel, which also apparently transports the major portion of $CO_2$. In combination, these processes ensure $NaHCO_3$ reabsorption. A small proportion of $H^+$ released by PCT cells into the lumen is saturated by various luminal buffers ($NH_3$, inorganic phosphate, creatinine) where both $NH_4^+$ excretion and acid formation can be measured (*Boron, 2006*). The transport processes in PCT are summarized in Fig. 2.

The regulation of AB balance in PCT involves several pH-sensitive molecules: calcium-sensing receptor CaSR, G-protein coupled receptor Gprc5c, receptor tyrosine kinases ErbB1/2, proline-rich tyrosine kinase Pyk2, pH-sensitive ion channel TASK2 and bicarbonate-stimulated soluble adenylyl cyclase (*Brown & Wagner, 2012*; *Rajkumar & Pluznick, 2018*). Pyk2 is a key mediator of increased NHE3 activity following acid stimulation. Pyk2 shows maximal kinase activity at low pH within the physiological range, decreasing from 7.4 to 7.0; experiments indicate that Pyk2 directly responds to the change in pH. Pyk2 knockdown with either dominant-negative *pyk2* or siRNA prevented the acid-induced NHE3 activation. The Pyk2 signaling cascade presumably involves endothelin receptor B (ETRB) known as NHE3 activity modulator (*Brown & Wagner, 2012*; *Rajkumar & Pluznick, 2018*). ETRB stimulation leads to RhoA-dependent cytoskeletal remodeling and NHE3 membrane accumulation followed by increased proton efflux from PCT cells (*Wesson, 2011*). Another pH regulation pathway includes ErbB1/2- and angiotensin II receptor type 1-mediated signaling cascade. ErbB1/2 is a putative $CO_2$/bicarbonate sensing molecule at the basolateral membrane of PCT cells (*Zhou et al., 2006a, 2006b*; *Skelton & Boron, 2015*), consistently with the observed activation of ErbB2 by mildly alkaline extracellular media (pH 8–9) (*Serova et al., 2019*). Analysis of ErbB1/2 tyrosine phosphorylation in rabbit PCT also indicates a response to acute AB imbalance (*Skelton & Boron, 2015*).

The proximal bicarbonate reabsorption is regulated by several hormones acting for either minutes-to-hours or days. The process can be stimulated by adrenergic agonists and angiotensin II, inhibited by parathyroid hormone *via* cAMP signaling and stimulated by hypercalcemia. In chronic acidosis, the binding of renal endothelin 1 to its receptor ETRB has been identified as the trigger of increased $Na^+/H^+$ exchange in PCT (*Wesson, 2011*). Glucocorticoids participate in the regulation as well, notably during the chronic response to acidosis (*Bobulescu et al., 2005*).

Of course, other ions, notably oxalate and phosphate, contribute to the maintenance of AB balance by the kidney as carriers. Expression of SLC26A6 at the apical membranes of PCT epithelium is crucial for NaCl reabsorption and facilitates oxalate secretion. SLC26A6 also plays a role in transtubular NaCl reabsorption and oxalate transport potentially involving SLC26A7 and AE1 at the basolateral membrane. In addition, SLC26A6 interacts with NaDC-1 to regulate citrate absorption from the urinary lumen thereby preventing

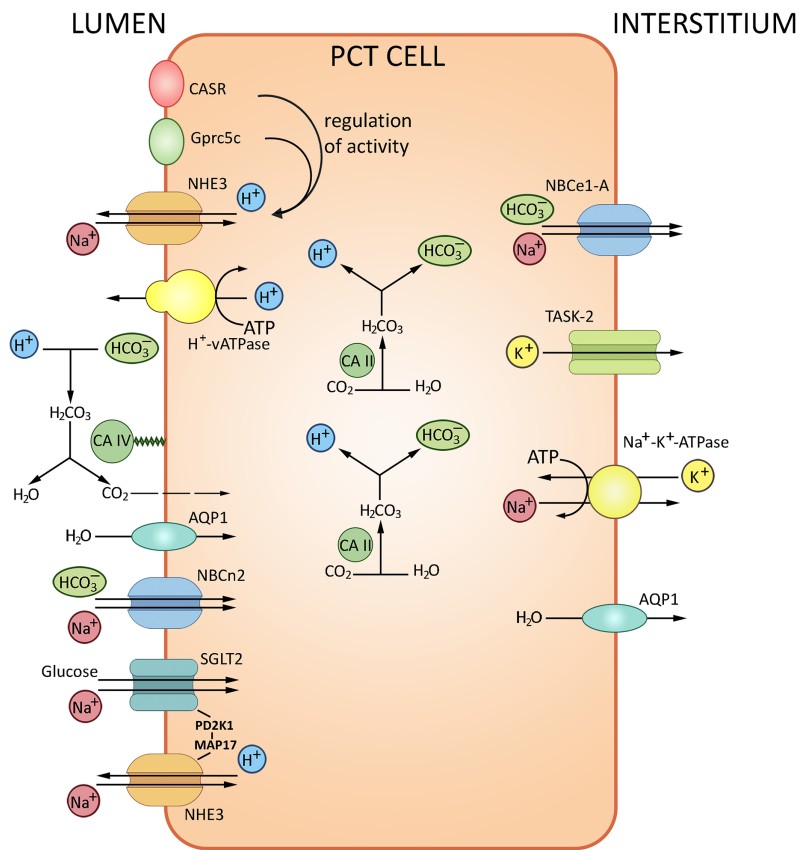

**Figure 2** **Routes of acid-base exchange in the proximal convoluted tubule (PCT), which reabsorbs the major portion of bicarbonate ($HCO_3^-$) from the glomerular filtrate.** The process involves chloride-bicarbonate exchange *via* sodium-dependent bicarbonate cotransporter at the luminal membrane of PCT cells. Protons ($H^+$) are actively secreted into the lumen in exchange for sodium ions ($Na^+$) *via* sodium-hydrogen exchanger NHE3 at the luminal membrane. PCT also reabsorbs filtered buffers such as phosphate ($HPO_4^{2-}$) and ammonia ($NH_3$) from the glomerular filtrate and participates in reabsorption and production of ammonium ($NH4^+$). NHE3, $Na^+/H^+$ exchanger 3; AQP1, aquaporin 1; CA II and CA IV, carbonic anhydrases II and IV; NBCe1-A, $Na^+/HCO_3^-$ electrogenic cotransporter 1, splice variant A, NBCn2, $Na^+/HCO_3$ Cotransporter 2, TASK2, two-pore domain $K^+$ Channel 2, SGLT2, $Na^+$/glucose cotransporter 2, CASR, calcium-sensing receptor, Gprc5c, G-protein coupled receptor family C group 5 member C. BioRender (biorender.com) was used for the creation of the figure.

$Ca^{2+}$ oxalate crystallization and kidney stone formation (*Wang et al., 2021*). Systemic levels of inorganic phosphate ($P_i$) are primarily adjusted by reabsorption in PCT involving SLC34A1 (NaPi-IIa, main), while SLC34A3 (NaPi-IIc, developmental) and SLC20A2 (PiT-2, accessory) $P_i$ transporters (*Burns & Werner, 2020*).

TASK2, another pH-sensitive molecule regulating bicarbonate reabsorption in PCT, is a two-pore domain $K^+$ channel activated by extracellular alkalization. TASK2-null mice show decreased water and sodium reabsorption in the kidneys under alkali load, leading to metabolic acidosis and significantly reduced arterial blood pressure compared to normal mice (*Warth et al., 2004*). TASK2 is coexpressed with $Na^+/HCO_3^-$ cotransporter NBCe1-A. The release of $Na^+$ and $HCO_3^-$ ions *via* NBCe1-A causes basolateral membrane depolarization. With the increase in extracellular $HCO_3^-$ concentration pH rises, activating

TASK2. The release of accumulated $K^+$ via TASK2 leads to repolarization of the membrane providing a driving force for $Na^+$ and $HCO_3^-$ reabsorption (*Warth et al., 2004*). Double inactivation of NBCe1-A and TASK2 results in metabolic acidosis (*Rajkumar & Pluznick, 2018*).

Soluble adenylyl cyclase (sAC) is expressed in kidney tubules and activated by $HCO_3^-$ ions; the response can be enhanced by calcium. The protein may function as a $CO_2$ sensor involved in cAMP-dependent V-ATPase membrane accumulation and proton secretion (*Levin et al., 2021*).

Calcium-sensing receptor CaSR is a G protein-coupled receptor (GPCR) expressed at the apical membrane. CaSR is sensitive to extracellular pH, with alkali-enhanced response to calcium and magnesium ions. CaSR stimulation increases NHE3 activity in PCT (*Rajkumar & Pluznick, 2018*). Gprc5c, also a GPCR at the apical membrane in PCT, is functionally similar to CaSR. Gprc5c knockout mice have abnormal pH homeostasis with lower blood pH, higher urine pH and reduced renal NHE3 activity compared to wild type, suggesting a role in regulation of systemic AB balance effectuated through NHE3 (*Rajkumar & Pluznick, 2018*).

Bicarbonate reabsorption in the kidney is also coupled to glucose reabsorption. A $Na^+$/glucose cotransporter SGLT2 (*Slc5a2*) is the main glucose transporter in the kidney. About 90% of glucose is reabsorbed in PCT via SGLT2 expressed at the apical membrane. SGLT2 appears to functionally interact with NHE3 at the apical membrane, possibly via PDZK1 and 17-kDa membrane-associated protein (MAP17). NHE3 mediates bicarbonate reabsorption and ammonia secretion and also contributes to $Na^+$ and water reabsorption. The SGLT2/NHE3 multiprotein complex may represent a functional unit that affords concomitant regulation of AB balance, extracellular fluid volume and glucose homeostasis. A nonspecific competitive SGLT inhibitor phlorizin markedly increased the urinary $Na^+$ and $HCO_3^-$ excretion (*Silva et al., 2020*). Selective SGLT2 inhibitors reduce both glucose reabsorption rates and blood pressure (*Burnier, Oe & Vallon, 2022*).

It should be mentioned that transport and metabolism of amino acids (AA) in PCT also contribute to AB balance regulation. The total amount of amino acids in the renal filtrate is reabsorbed in PCT and partially metabolized (mainly glutamine) producing ammonia and bicarbonate. Glutamine transport and metabolism are coupled to proton excretion (*Harris et al., 2023*). The glutamine transfer by apical $Na^+$-dependent neutral amino acid transporter 1 and basolateral $Na^+$-coupled neutral amino acid transporter 3 is complemented by glutaminase activity. In acidosis, glutaminase activity is significantly enhanced to afford the clearance of $H^+$ as $NH_4^+$ (*Li, Zheng & Wu, 2020*).

The proximal reabsorption of bicarbonate is extensively regulated; however, residual amounts of this ion are efficiently reabsorbed in the distal PCT through different regulatory mechanisms.

### Loop of Henle

The proximal convoluted tubule continues into the loop of Henle, a U-shaped epithelial tubule running into the medulla and returning to the cortex where it continues into the distal convoluted tubule of the nephron. The loop of Henle consists of two limbs—a

descending and an ascending, connected by a thin-walled middle segment much narrower than other renal tubules (*Pallone, Zhang & Rhinehart, 2003*). The descending limb does not participate in ion transport: it is only permeable to water, which flows through AQP1 and AQP3 channels in the apical and basolateral membranes of the descending thin tubule; the driving force is provided by small constant osmotic gradients of solutes maintained locally (*Agre et al., 2002*).

The thick ascending limb of the loop of Henle (TAL) is largely responsible for the water content of the urine: from highly concentrated small volumes of urine in anti-diuresis to large volumes of very low osmolarity in water diuresis. Specifically, TAL is responsible for diluting the content of the lumen while mounting the osmotic pressure in the interstitium. TAL is impermeable to water; the dilution occurs through active reabsorption of NaCl which feeds the hypertonic gradient. The resulting interstitial hypertonicity of the medulla provides a driving force for water reabsorption from the urine at the final stages of its formation in collecting tubules of the medulla. In water diuresis, collecting tubules become impermeable to water and the urine stays diluted (*Greger, 1985*).

The loop of Henle (particularly TAL) reabsorbs about 40% of $Na^+$ present in the glomerular filtrate. Sodium ions enter cytosol mostly by $Na^+/K^+/2Cl^-$ electroneutral cotransporter NKCC2 (encoded by *SLC12A1*) at the apical cell membrane. Unlike its closely related isoform NKCC1 (*SLC12A2*) expressed in multiple organs and tissues, NKCC2 is a TAL-specific protein (*Zacchia et al., 2018*). Sodium ions subsequently leave the cells by sodium pumps at the basolateral surface; the parallel basolateral efflux of chloride ions is mediated by ClC-Ka and ClC-Kb channels with Barttin subunits (*Hennings et al., 2017*). By contrast, potassium ions flow back into the lumen *via* the renal outer medullary $K^+$ (ROMK) channels thereby recovering the gradient required for the sodium-chloride symport and simultaneously the positive potential on the apical membrane which coordinates the paracellular reabsorption of cations (*Zacchia et al., 2016*).

The combined activity of essential transporters and ion channels involved in the salt reabsorption (NKCC2, ROMK, ClC-Kb) ensures the electrolyte balance. A defect in any of these components may cause a salt-wasting phenotype. However, as the reabsorption in TAL is regulated by hormones and paracrine factors through multiple intracellular signaling pathways, renal salt wasting often reflects abnormal regulation of the non-affected transport machinery. Moreover, activating mutations in transporter-encoding genes can ultimately lead to salt-wasting nephropathy, which illustrates the complex nature of the water-salt homeostasis. Defects in uromodulin, Ste20-related proline alanine rich kinase (SPAK) and oxidative stress responsive kinase (OSR1) molecules may affect water-salt homeostasis by influencing NKCC2 activity (*Watanabe et al., 2002*).

The loop of Henle contributes to AB homeostasis by reabsorbing bicarbonate (up to 15% of the initial content in the glomerular filtrate). In rats, the descending limb is fairly permeable to bicarbonate; nevertheless, its concentration inside the loop decreases toward the U-turn, reflecting the high-rate water reabsorption (*Capasso et al., 1991*). In TAL, bicarbonate is reabsorbed transcellularly; the mechanisms (NHE-mediated exchange) are

similar to those in PCT. Of NHE2 and NHE3 isoforms found in the membranes, NHE3 is essential to AB homeostasis as indicated by targeted inactivation of corresponding genes in mice (*Watts & Good, 1994*; *Ledoussal et al., 2001*). *In vivo* and *ex vivo* perfusion experiments show that bicarbonate reabsorption requires carbonic anhydrase and can be stimulated by bumetanide. The bicarbonate efflux from cells is mediated by $Cl^-/HCO_3^-$ exchanger 2 (AE2). A similar exchanger, AE1, is expressed at the apical (luminal) surface and coupled to $Na^+/H^+$ activity thereby particiapting in $Na^+$ reabsorption (*Eladari et al., 1998*; *Houillier & Bourgeois, 2012*). The major contribution of NHE3 is complemented by several accessory transporters. Functional and immunohistochemical tests reveal $H^+$ ATPase expression along TAL. Its role in bicarbonate reabsorption is considered secondary, under physiological conditions often superfluous, but still significant for AB homeostasis. Shifts in AB balance modulate the rates of bicarbonate reabsorption along TAL: the process is stimulated in metabolic acidosis and suppressed in alkalosis, independently of the acute or chronic nature of the condition. According to functional studies, both NHE and $H^+$-ATPase activities adapt to changes in AB status, apparently under hormonal stimulation (glucocorticoids, aldosterone) (*Capasso et al., 1994*). Another phosphate transporter, sodium-dependent SLC34A2 co-localized with uromodulin in TAL, is also active in distal renal tubules (*Burns & Werner, 2020*).

$NH_3$ trafficking is another important regulatory factor of ion exchange in the loop of Henle. The bulk of excreted ammonium is absent in the glomerular filtrate, but produced from glutamine in the kidney and secreted to the urine (*Weiner & Verlander, 2013*). TAL plays a decisive role in $NH_3$ reabsorption; the process involves the NKCC2-mediated potassium binding. The $K^+/NH_4^+$ exchange and $K^+$ transfer in TAL have been described as well, albeit provide a minor contribution compared to the NKCC2-mediated paracellular effects. The basolateral efflux of cations in TAL depends on NHE4 ($Na^+/NH_4^+$), NBCn1 ($Na^+/HCO_3^-$) and chloride transporters (*Blanchard et al., 1998*). The routes of ion exchange in the loop of Henle are summarized in Fig. 3.

### Distal convoluted tubule

The loop of Henle continues into the distal convoluted tubule—a relatively short portion of the nephron critically important for sodium, potassium and divalent cation homeostasis. DCT also participates in NHE2-dependent bicarbonate reabsorption. DCT cells are rich in mitochondria supporting the electrolyte transporters, notably the basolateral $Na^+/K^+$-ATPase. The basolateral efflux of bicarbonate proceeds by AE2-mediated exchange with chloride, but may also involve basolateral chloride channels partially permeable to $HCO_3^-$. The cells express cytosolic carbonic anhydrase CA II, but lack the membrane-bound CA IV at the apical surface (*Subramanya & Ellison, 2014*). The electrolyte trafficking in DCT involves sodium, potassium and chloride ions. Sodium transport in DCT (up to 10% of the overall $Na^+$ reabsorption) is mediated by thiazide-sensitive NaCl cotransporter (NCC, *SLC12A3*); sodium-dependent $Cl^-/HCO_3^-$ exchanger (NDCBE, *SLC4A8*) and amiloride-sensitive epithelial sodium channel (ENaC) (*Leviel et al., 2010*); NCC is regulated by a complex cascade of WNK-dependent serine-threonine kinases SPAK and OCR-1. Chloride transport uses ClC-Kb channels and KCl cotransporter 4 (KCC4,

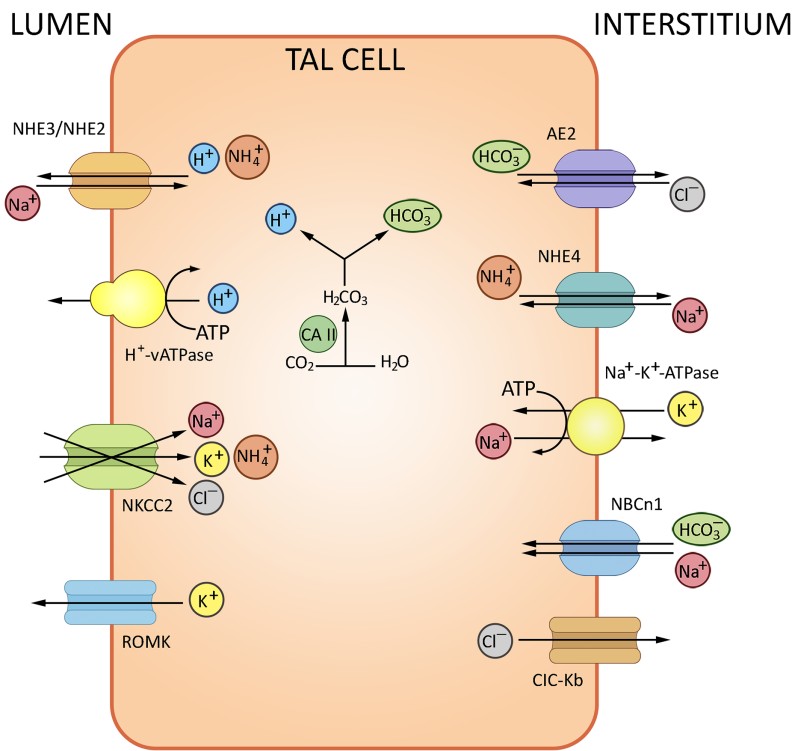

**Figure 3 Routes of acid-base exchange in the thick ascending limb of the loop of Henle (TAL).**
Reabsorption of sodium and chloride in TAL indirectly affects bicarbonate concentration in the renal tubular fluid. TAL actively reabsorbs sodium and chloride ions *via* specific transporters, notably the Na-K-2Cl cotransporter NKCC2, contributing to the osmotic gradient that drives the reabsorption of water in the collecting duct. TAL contributes to AB balance by indirectly influencing the delivery of sodium and bicarbonate to distal nephron segments where final adjustments occur. NHE2/NHE3, Na$^+$/H$^+$ exchangers 2 and 3; NKCC2, Na$^+$/K$^+$/2Cl$^-$ cotransporter; AE2, Cl$^-$/HCO$_3^-$ antiporter, ROMK, renal outer medullary K$^+$ channel, NCBn1, Na$^+$/HCO$_3^-$ Cotransporter 1, NHE4, Na$^+$/H$^+$ Exchanger 4, ClC-Kb Cl$^-$ Channel Kb. BioRender (biorender.com) was used to make the figure.

*SLC12A7*). Potassium secretion in DCT involves the 'big K$^+$' and ROMK channels (*Meneton, Loffing & Warnock, 2004*). The routes of ion transport in DCT are summarized in Fig. 4.

### Collecting tubules and collecting ducts

The epithelial collecting system of the kidney connects nephrons to the urinary tract. This system substantively contributes to the water-electrolyte balance by means of aldosterone/ADH-dependent reabsorption and secretion (*Olde Engberink et al., 2023*). The walls of collecting tubules comprise two types of epithelial cells, principal cells (PC) and intercalated cells (IC) distinguished morphologically and expressing specific sets of ion channels and transporters (Fig. 5). PC express the epithelial Na$^+$ channel ENaC and aquaporin AQP2 at the apical membrane, and AQP3/AQP4 at the basolateral membrane. AQP2 marks the entire length of the epithelial collecting system, from connecting tubules through cortical and outer medullary collecting tubules to inner medullary collecting tubules and ducts. IC, which express H$^+$-ATPase V and carbonic anhydrase CAII, are

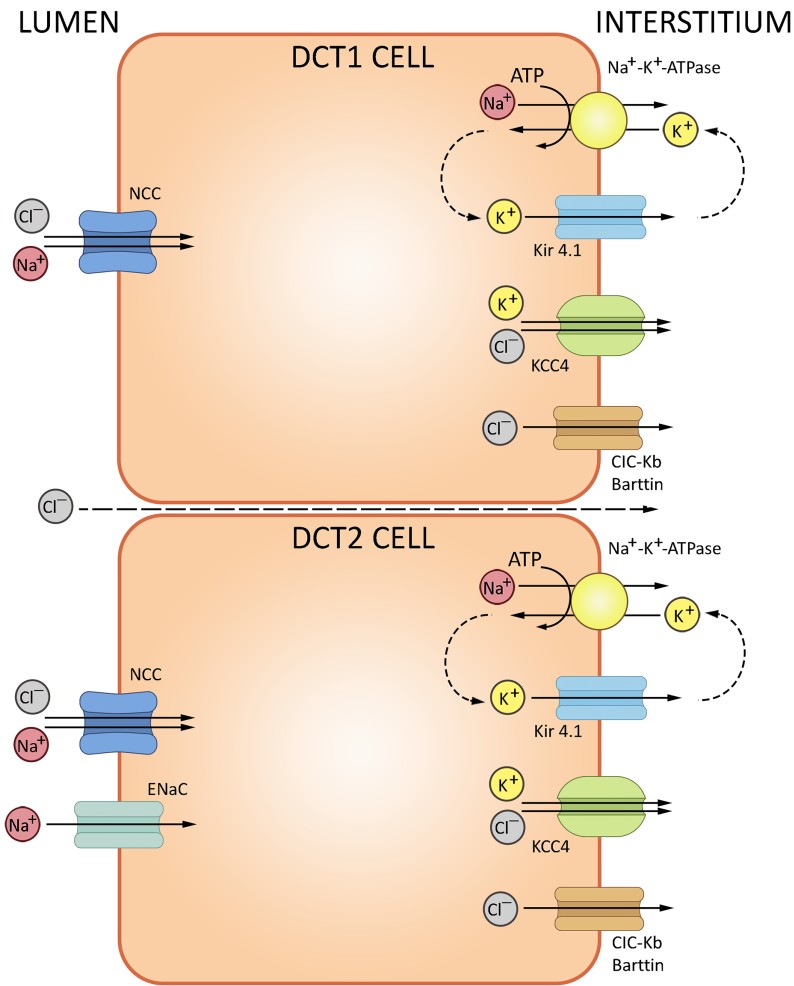

**Figure 4 Routes of acid-base exchange in the distal convoluted tubule (DCT).** Secretion of hydrogen ions (H⁺) to the lumen proceeds *via* $H^+$-ATPase pump located at the apical membrane. DCT also participates in bicarbonate ($HCO_3^-$) reabsorption from the tubular fluid *via* $Cl^-/HCO_3^-$ exchanger, excretion of ammonium ($NH4^+$), reabsorption of sodium and secretion of potassium ions. NCC, NaCl cotransporter; ENaC, epithelial sodium channel; ClC-Kb, chloride channel; KCC4, potassium/chloride cotransporter. BioRender (biorender.com) was used for the creation of the figure.

engaged in the acid/bicarbonate transport: V-ATPases acidify certain cytoplasmic compartments and pump protons across plasma membranes, while carbonic anhydrases catalyze the production of $H^+$ and $HCO_3^-$ from carbon dioxide and water. Intercalated cells of the collecting system can be further subdivided into IC-α and IC-β based on molecular markers and functional differences specified in Table 1.

IC-α express $H^+$-ATPase V at the apical membrane and $Cl^-/HCO_3^-$ anion exchanger at the basolateral membrane, responsible for proton secretion and bicarbonate reabsorption, respectively. ATPase V is regulated *via* soluble adenylate cyclase cascade. Electrogenic chloride transporter A11 (*SLC26A11*) is also expressed at the apical membranes of IC-α (*Roy, Al-Bataineh & Pastor-Soler, 2015a*). The acidified urine is less susceptible to microbial colonization. Moreover, though antimicrobial defense cannot be regarded as a

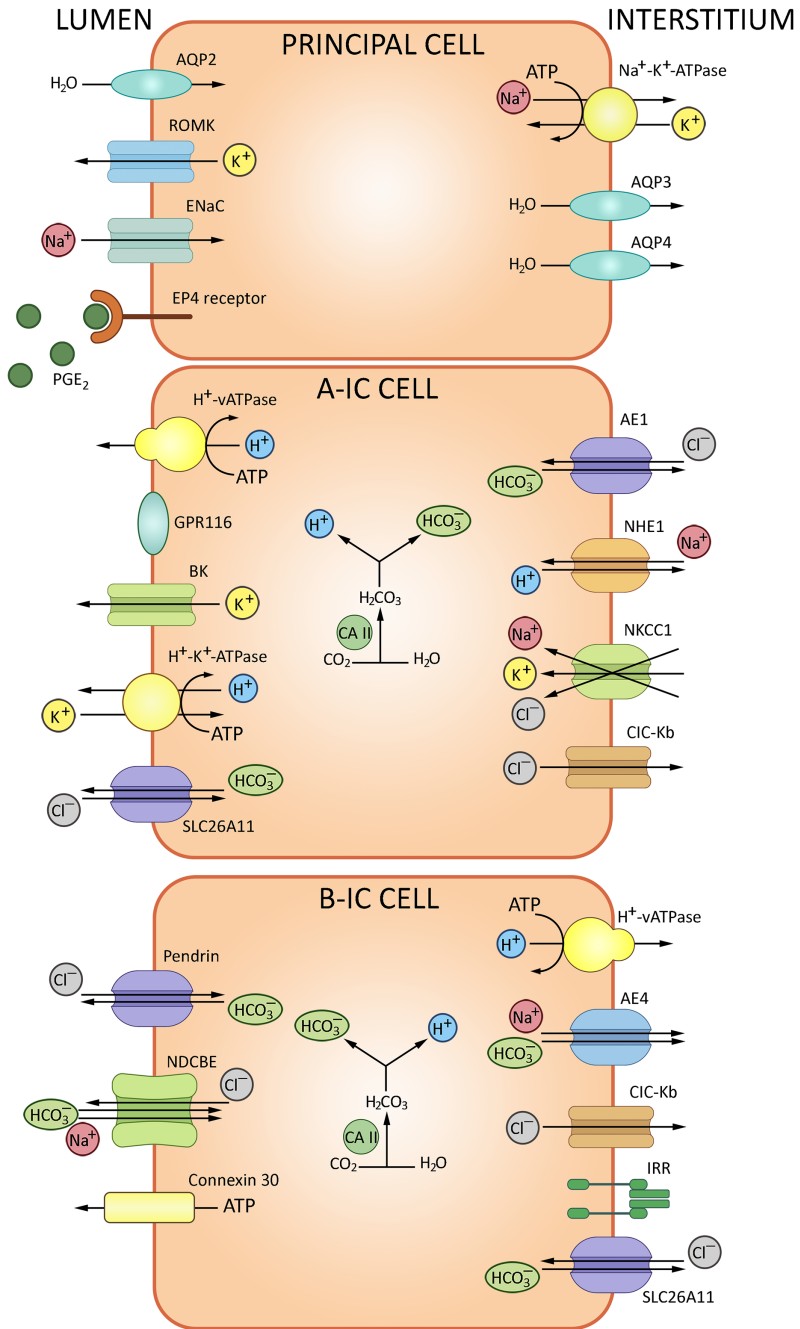

**Figure 5 The routes of acid-base exchange in cortical collecting tubules.** ATPase V apically and Cl⁻/
HCO₃⁻ anion exchanger basolaterally, ensuring proton secretion to the urine and bicarbonate reabsorption, respectively, whereas intercalated cells β express pendrin for bicarbonate secretion. A minor subpopulation of intercalated cells express both pendrin and H⁺-ATPase at the apical membrane and apparently secrete both bicarbonate and protons. Overall, secretion of protons and reabsorption of bicarbonate involves a complex interplay of transporters and channels including proton pumps and H⁺/ ATPase to maintain systemic pH balance. BioRender (biorender.com) was used for the creation of the figure.

**Table 1 Functional profiles of intercalated epithelial cells of the collecting tubules of the kidney.**

| Cell type | Secrete | Reabsorb | Exchangers characteristic |
|---|---|---|---|
| Intercalated type α | Acid = H⁺ ions; | Bicarbonate | Apical H$^+$-ATPase V, electrogenic chloride transporter A11, basolateral anion exchanger AE1 Cl$^-$/HCO3$^-$, H$^+$/K$^+$ exchanger NKCC1, sodium-hydrogen antiporter 1 NHE1 |
| Intercalated type β | Bicarbonate | Acid | Apical Pendrin, sodium-dependent Cl$^-$/HCO$_3^-$ exchanger (NDCBE), basal H$^+$ V-ATPase and anion exchanger AE4 |
| Intercalated non-α non-β | Acid and bicarbonate | – | Apical H$^+$ ATPase, H$^+$/K$^+$ exchanger, pendrin (Slc26a4) |

direct function of IC-α, these cells participate in the innate immunity reactions by producing lipocalin, a neutrophil gelatinase-associated protein that binds and inhibits siderophores—small chelating compounds secreted by bacteria and fungi to promote iron transfer across cell membranes (*Paragas et al., 2014*).

IC-β are marked by apical expression of pendrin SLC26A4, which mediates bicarbonate secretion, and basolateral expression of V-ATPase engaged in H$^+$ reabsorption and sodium-dependent Cl$^-$/HCO$_3^-$ exchanger NDCBE (*Eladari et al., 1998*; *Petrenko et al., 2013*). Apart from pendrin, the base excretion involves cystic fibrosis transmembrane conductance regulator (CFTR); both proteins are found at the apical membrane of IC-β. In human patients and experimental mice with cystic fibrosis this functionality is impared. CFTR-dependent transport is crucial for HCO$_3^-$ secretion to the urine (*Berg et al., 2021*). Anoctamins ANO1/6 are required for proper expression and function of CFTR. Thus, in renal collecting system, bicarbonate secretion occurs through a synergistic action of CFTR and pendrin supported by ANO (*Kunzelmann et al., 2023*). Anion exchanger 4 (AE4), also implicated in bicarbonate transport, is specifically expressed at the basolateral membrane of IC-β. Although sodium reabsorption *via* AE4 is too weak for plasma volume regulation, the response of IC-β to AB-related metabolic challenges depends on AE4 (*Vitzthum, Meyer-Schwesinger & Ehmke, 2024*).

Conventionally, major cell types of the collecting system were assigned distinct functionalities, with IC-α and -β ensuring, respectively, acid and base secretion and principal cells reabsorbing sodium and water and secreting potassium. However, recent studies reveal broader functional profiles. In particular, IC-β are capable of NaCl absorption and participate in regulation of extracellular fluid volume and blood pressure. In β-ICs, the apical influx of Cl$^-$ *via* pendrin is coupled to the apical Na$^+$ influx *via* NDCBE (*Almomani et al., 2014*; *Roy, Al-Bataineh & Pastor-Soler, 2015b*). An increasing amount of evidence indicates that IC control the maintenance of Na$^+$ balance by adjacent principal cells. In particular, Na$^+$ absorption *via* ENaC in principal cells is coordinated to Cl$^-$ absorption *via* pendrin in IC to operate the NaCl absorption (*Wall, Verlander & Romero, 2020*), while luminal bicarbonate levels have been positively associated with ENaC expression and activity. Therefore, pendrin can modulate ENaC; the link is mediated by luminal HCO$_3^-$ concentrations or pH. Pendrin gene ablation leads to decreased abundance

of $H^+$-ATPase in IC-$\beta$ and a concomitant increase in the intracellular ATP levels. The luminal ATP levels subsequently rise due to enhanced ATP secretion *via* connexin 30 which contains a $CO_2$ binding site. When $pCO_2$ rises, the hemichannel opens and releases ATP (*Serova et al., 2020*). In the lumen, ATP binds purinergic receptors at the apical surface to promote calcium-dependent production of prostaglandin E2 which downregulates ENaC in principal cells. This loop links bicarbonate transport to regulation of sodium balance, extracellular fluid volume and blood pressure (*Gueutin et al., 2013*). Another signaling molecule expressed by IC-$\beta$, the insulin-receptor related receptor (IRR), participates in bicarbonate excretion regulation and provides a pH-sensor of extracellular fluid alkalinization (*Gantsova et al., 2022*; *Korotkova et al., 2022*). In mice, IC-$\alpha$ of cortical collecting ducts can secrete $Na^+$ and $Cl^-$ *via* NKCC1, probably HKA2 and a $Cl^-$ channel (*Morla et al., 2016*). Recent studies indicate that principal cells are involved in the control of AB homeostasis. For instance, high systemic acid loads stimulate ADH production. Subsequent activation of Avpr2 receptor for ADH in principal cells triggers production of Gdf15 which activates the ErbB2 signaling pathway in IC-$\alpha$ to induce their proliferation (*Cheval et al., 2021*).

A minor subpopulation of IC is neither $\alpha$, nor $\beta$ (Table 1); these cells express both pendrin and $H^+$-ATPase in the apical membrane and apparently secrete both bicarbonate and protons. IC have more restricted representation in the collecting system than principal cells. IC-$\alpha$ are found in connecting tubules, cortical collecting tubules and initial portions of medullary collecting tubules. IC-$\beta$ are found in connecting tubules and cortical collecting tubules, but typically absent in the medullary collecting tubules. PC and IC also express alternative $Cl^-$/$HCO3^-$ exchangers (*SLC26A7*, *SLC26A11*), AE4 (*SLC4A9*) and $Na^+$/$K^+$-ATPase (*Chen, Higgins & Zhang, 2017*).

Another G-protein-coupled receptor implicated in AB homeostasis, Gpr116, is expressed in cortical collecting ducts by IC-$\alpha$. Tubule-specific Gpr116 inactivation in mice leads to production of acidic urine and mild systemic alkalosis; the analysis reveals impaired distribution of V-ATPase proton pumps at the surface of IC-$\alpha$. Gpr4, another acid-sensing molecule expressed in collecting tubules, is significant as well: genetic inactivation of Gpr4 reduces net acid secretion leading to acidosis (*Zaidman & Pluznick, 2022*).

The epithelial collecting system also implements $NH_3$/$NH_4^+$ transport regulated by a complex process that involves at least two Rhesus glycoproteins: RhB (SLC42A2) and RhC (SLC42A3) (*Harris et al., 2023*; *Takvam et al., 2023*). Rhbg expressed in IC-$\alpha$ and non-$\alpha$, non-$\beta$ intercalated cells, principal cells in the basolateral plasma membrane. Rhbd is not found in IC-$\beta$. Rhcg is found in the apical and basolateral membrane in type IC-$\alpha$ and principal cells, but also it is expressed in the apical plasma membrane in non-$\alpha$, non-$\beta$ intercalated cells. Recent reports show that RhC is specifically involved in transport of the molecular form ($NH_3$), whereas RhB participates in transport of both $NH_3$ and $NH_4^+$ (*Weiner & Verlander, 2014*). Collecting duct-specific double knockout of these proteins confirms the role of RhB/C in kidney response to metabolic acidosis. Compared with wild type mice, mice with collecting duct Rhbg and Rhcg deletion had significantly more severe metabolic acidosis more severe HCl-induced metabolic acidosis (*Lee et al., 2014*).

## Clinical manifestations of acid-base metabolism disorders

Clinical manifestations of changes in AB balance at systemic level, directly or indirectly related to kidney function, include excessive acid production; acid wasting; excessive production/retention of bicarbonate; and bicarbonate wasting.

*Excessive acid production* is observed in diabetic ketoacidosis. The insulin deficiency stimulates the release of free fatty acids from adipose tissue, subsequently oxidized to ketone bodies—acidic compounds used for energy production in diabetes. Ketone bodies, notably acetoacetate and β-hydroxybutyrate, overload the bicarbonate buffer system causing metabolic acidosis manifested as decreased pH of the blood (*Sheikh-Ali et al., 2008*). Given the pathogenesis of this disorder, it is necessary, first of all, to prescribe or correct insulin therapy, as well as appropriate infusion therapy. Patients are administered physiologic solution, and if the pH drops below 6.9, sodium bicorbanate is administered (*Gosmanov, Gosmanova & Dillard-Cannon, 2014*).

*Acid wasting.* Pyloric obstruction (*e.g.*, in stenosis) blocks the passage of gastric content to the duodenum. The patients develop dehydration and vomiting; the massive loss of hydrochloric acid from the stomach may cause hypochloremic metabolic alkalosis, notably in infants (*van den Bunder et al., 2020*). The treatment options include arginine hydrochloride administration; the doses depend on the patient's weight and clinical laboratory parameters (*Sierra, Hernandez & Parbuoni, 2018*). Carboanhydrase inhibitor acetazolamide is considered a safe and efficacious option for metabolic alkalosis in pediatric cardiology (*Moffett, Moffett & Dickerson, 2007*).

*Excessive production/retention of bicarbonate.* Patients with mixed or type 2 respiratory failure often develop metabolic alkalosis, with increased bicarbonate levels compensating for the rise in blood $CO_2$. Such post-hypercapnic alkalosis interferes with the respiratory function recovery *via* autonomic circuits (*Yi, 2023*). Acetazolamide, a carbonic anhydrase inhibitor with mild diuretic effect, increases the rates of bicarbonate excretion with urine, thereby alleviating the systemic alkalosis (*Gulsvik et al., 2013*). The overall beneficial effect allows a transfer from artificial ventilation to spontaneous breathing.

*Bicarbonate wasting.* Alpinists at high altitudes experience a switch to hyperventilation breathing mode in response to the low oxygen pressures. The rapid breathing leads to hypocapnia and respiratory alkalosis preventing a further compensatory increase in the ventilation volume *via* autonomic circuits. Eventually, increased excretion of bicarbonate with the urine restores the systemic pH balance. The adaptation can be supported by acetazolamide administration which accelerates pH adjustment to the normal range by facilitating bicarbonate excretion (*Tanios et al., 2018*; *Clayton-smith, 2021*).

It is important to consider genetic basis of particular failures in AB homeostasis, notably mutations that affect bicarbonate transport in the kidney. Mutations in SLC26A1 and SLC26A6 have been associated with defective bicarbonate reabsorption in PCT, distal renal tubular acidosis and urinary bicarbonate wasting, though the mechanism remains unknown (*Wang et al., 2021*; *Li et al., 2023*). Cystic fibrosis is an inherited condition resulting from mutations in CFTR affect bicarbonate secretion in the cortical collecting system as a part of cystic fibrosis phenotype. The dysregulated bicarbonate excretion by the

kidneys may result in systemic metabolic alkalosis (*Kunzelmann et al., 2023*). The $Ca^{2+}$-activated $Cl^-$ channel TMEM16A and aquaporin 2 play central roles in the autosomal dominant polycystic kidney disease (ADPKD) (*Talbi et al., 2021*; *Olesen & Fenton, 2021*). SLC4A1 dysregulation has been linked to distal renal tubular acidosis and mutant SLC4A2, SLC4A4 and SLC4A5 have been implicated in impaired proximal bicarbonate reabsorption (*Du et al., 2021*; *Zhong et al., 2023*). Mutations in SLC34A1 have been associated with nephrocalcinosis, nephrolithiasis, hyperoxaluria and hypercalciuria (*Howles & Thakker, 2020*). Mutated proton pumps ATP6V1B1 and ATP6V0A4 have been associated with distal renal tubular acidosis (RTA) in nephrocalcinosis and nephrolithiasis. Mutations in carbonic anhydrase II (CAII) lead to mixed proximal and distal RTA (type III RTA) of varying severity, often associated with nephrocalcinosis and nephrolithiasis (*Eaton, Merkulova & Brown, 2021*; *Alexander & Bockenhauer, 2023*).

COVID-19 posed a grave challenge to healthcare at the global level and pathophysiological aspects of COVID-19 remain a close focus. Altered electrolyte metabolism in patients with COVID-19 has been associate with the severity and mortality of the disease (*Lippi, South & Henry, 2020*; *Tezcan et al., 2020*). A recent study by *Jiang (2022)* reveals 100% incidence of metabolic alkalosis in a cohort of critically ill patients with mortality rate of 81.3% compared to 50% incidence in patients with severe form of the disease and mortality rate of 21.4%. *Al-Azzam et al. (2023)* also demonstrate a significant association of altered AB homeostasis with mortality rates among patients diagnosed with COVID-19, with significantly higher risks of lethal outcome in patients with metabolic acidosis associated with respiratory compensation, respiratory alkalosis with metabolic compensation and even patients with uncompensated respiratory acidosis. *Mansouri et al. (2022)* observed AB imbalances in 87.7% of patients hospitalized with COVID-19, particularly respiratory alkalosis in 22.4% of the cases. A number of other studies feature respiratory alkalosis as a common complication in COVID-19, associated with higher risks of severe clinical manifestations and definitely an indication for intensive monitoring. On the other hand, given the diversity of AB imbalances in COVID-19, it is difficult to propose a universal correction protocol for respiratory support combined to infusion therapy (*Alfano et al., 2022*).

## CONCLUSION

The acid-base homeostasis is fundamental for health as it ensures the optimal functioning of cells and biomolecules. The adjustment of bicarbonate and hydrogen content of body fluids by the kidneys is controlled by a complex regulatory network and provides significant resistance to metabolic shifts in acidity. In this review, we examined the main regulatory mechanisms that are carried out in the kidney to maintain the balance of acids and alkalis.

Changes in pH affect many processes in the body, both normal and pathological, with the kidneys playing a significant role in this regard. Understanding of the mechanisms of pH-sensitivity, as well as the influence of $H^+$ ion concentration on intracellular signaling, inflammation and immunogenesis processes will allow us to develop new ways of

treatment, first of all, of chronic renal failure. Alkali therapy may serve as one of such directions, so as pH-sensitive proteins could be a target to novel investigation.

Limitations of the study: The focus of the review primarily lies on the acid-base exchange in kidney and it is crucial to note that the effects of acid-base regulation through the renal pathway manifest over an extended period, often taking days to become apparent. Acid-base homeostasis maintained through the respiratory pathway is more rapid. The swiftness with which the respiratory system responds to changes in pH contributes significantly to maintaining equilibrium in the short term. The review, due to its scope and focus, may not comprehensively capture the nuances of the renal pathway's contribution within the confines of a shorter time frame.

### Funding
The corresponding results were obtained with the financial support of the Russian Federation represented by the Ministry of Education and Science of Russia; Agreement dated October 7, 2021 No. 075-15-2021-1356 (internal number of the Agreement 15. SIN.21.0011); (ID: RF 0951.61321X0012). The part of work concerning clinical manifestation was supported by the state task No. 122030200537-5 "Study of the mechanisms of activation of resident liver macrophages as a key stage of reparative morphogenesis". The part of work concerning kidney metabolism was supported by Russian Science Foundation (Grant Number 22-15-00241). The funders had no role in study design, data collection and analysis, decision to publish, or preparation of the manuscript.

### Grant Disclosures
The following grant information was disclosed by the authors:
Russian Federation represented by the Ministry of Education and Science of Russia; Agreement dated October 7, 2021: 075-15-2021-1356, 15.SIN.21.0011, RF 0951.61321X0012.
State Task: 122030200537-5.
Russian Science Foundation: 22-15-00241.

### Competing Interests
The authors declare that they have no competing interests.

### Author Contributions
- Elena Gantsova performed the experiments, analyzed the data, prepared figures and/or tables, and approved the final draft.
- Oxana Serova performed the experiments, prepared figures and/or tables, and approved the final draft.
- Polina Vishnyakova performed the experiments, analyzed the data, prepared figures and/or tables, and approved the final draft.
- Igor Deyev conceived and designed the experiments, authored or reviewed drafts of the article, and approved the final draft.
- Andrey Elchaninov conceived and designed the experiments, authored or reviewed drafts of the article, and approved the final draft.
- Timur Fatkhudinov conceived and designed the experiments, authored or reviewed drafts of the article, and approved the final draft.

## Data Availability

This is a literature review.

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
