# Peer review of "Mechanisms and physiological relevance of acid-base exchange in functional units of the kidney"

_PeerJ, doi:10.7717/peerj.17316_

## Round 0.1 · original submission · Major Revisions

Please respond to all the reviewer's comments and resubmit the revised manuscript with track changes. Along with that also submit point by point response to reviewers.

**Language Note:** The review process has identified that the English language must be improved. PeerJ can provide language editing services - please contact us at [email protected] for pricing (be sure to provide your manuscript number and title). Alternatively, you should make your own arrangements to improve the language quality and provide details in your response letter. – PeerJ Staff

·

Basic reporting

The review is clear in explaining the underlying fundamental mechanisms of AB exchange and balance in the kidneys. However, the manuscript could include more literature citations at relevant places in the text.

Experimental design

The review could include sub-headings or sections to provide more clarity for readers to understand the various sections. Also, a section on how AB balance affects various diseases should be clearly separated.

If possible, the authors could also include the baseline concentrations of various ions such as HCO3-, NH4+, H+, SO42- etc... (in plasma, various tissues, etc.) and how these change with various physiological states, especially in the context of diseases.

Also, the abbreviations should all be expanded at their first instance of mention in the text wherever possible

Validity of the findings

The findings of this review would be useful in primarily understanding the effects of AB homeostasis and its dysregulation in various kidney diseases and more broadly as a secondary cause or affect in other diseases such as COVID-19, as mentioned in the manuscript.

Additional comments

The authors could consider adding a "Limitations of the Study" section as they mention that the majority of AB homeostasis is maintained through the respiratory pathway and is more rapid. The effects of AB regulation via the renal pathway manifest over days, and this could be discussed in the proposed section in slightly more detail as well.

Reviewer 2 ·

Basic reporting

Basic reporting:

Gantsova et al presents an overview of the mechanistic and physiological relevance of acid-base exchange in the kidney. For the most part I enjoyed reading the review paper which is well written. However, given the detailed description on how the authors have conducted their literature research I am surprised that several crucial transport categories are missing in this review. Many of the described transporters in the review have been associated with severe kidney disease and acid-base disturbances (nephrolithiasis, kidney stones etc.) which to my surprised have received very limited attention by the authors. Please see more information on study design for improvements.

The description of the figure legends is not sufficient. Using 2-3 lines in the figure text to describe complex transport routes is not adequate. Please describe in detail the transport processes that are illustrated and the function of the transporters. A figure and a figure legend should be understood independently of the main texts. At present this is not possible. In addition, the authors have no references in the figure legends. You need to credit the research in which you have based your figures on. Please revise.

Although this is a review focusing on the mammalian kidney, I have included a few recent papers (both reviews and research papers) on other vertebrate species (zebrafish, medaka etc) which are commonly used animal models to study kidney physiology. Comparative nephron segments include proximal tubules, distal and collecting tubules. I hope the authors find these helpful. I do understand the more specific focus on bicarbonate regulation (which I have no problem with), but the authors introduce several other topics such as ammonia regulation, salt and water regulation where several important transport mechanisms are completely missing.

There are other recent reviews with similar focus as the current paper and I ask the authors to give a good explanation to why this paper is different. I have given several references for more recent papers in the field which have not been included by the authors. The review, although well written, need major revisions to be accepted for publication. If the authors are not able to meet these standards I do not support publication of this review article.

Experimental design

I really question the validity of the literature research by the authors as there are many recent papers on the topic not included by the authors and to be honest many references are very old. I count only about 18-20 (out of a total of 70) references that are published after 2015 and several of these are not related to acid-base exchange. See my suggestions of papers published between 2017-2023. I encourage the authors to use these papers to find more updated literature. It is okey to have older papers (currently 70-75 % of references are older than 10 years), but this needs to be balanced out with the more recent advances in the field. This is particular important writing review articles where the reader expect that the authors provide them with the most updated literature. This is not the case in the current version.
The reference list is in absolute terrible shape. There are many errors: missing the year, missing the journal name, missing volume, major spelling errors, different reference styles etc. Please look through the reference in detail and the specific comments below. Here are just a some errors I noticed:

Line 295: missing year on Capasso et al citation

Line 328: missing year on Olde Engberink et al., citations

Line 456: missing year on the Capasso et al in the reference list

Line 501: Kriz and Kaissling, 2008 are missing journal name and volume. Please fix

Line 503: Kurtzman NA are missing year, journal name and volume.

Line 518: Mansouri et al., 2022 are missing journal name and volume

Line 527: Nagami 1989 are missing journal name and volume

Line 529: missing year on the Olde Engbering et al in the reference list

Line 533: Authors names are not correct or misspelled

Line 534: Pallone et al., 2003 missing journal name and volume

Line 549: Pitts et al 1946 missing journal name and volume. Several misspellings

Please see the following improvements and what is missing in this review:

1. Reviewing mechanism and physiological relevance of acid-base exchange is a daunting task especially targeting specific functional units of the kidney. Although you describe several important transporter families involved in acid-base regulation in the kidney you are missing crucial transport categories, even transporters involved in bicarbonate transport, which is your main focus. This includes members within the SLC26 family referred to as the SLC26A6. I am surprised that the authors missed this as it is the member within the SLC26 family with most extensive transport function that can mediate Cl-/HCO3- transport and have been located apically of the kidney proximal tubules. Describing the role of the SLC26A1 could also benefit the review as it considered to mediate basolateral oxolate-SO42--/HCO3- exchange in the proximal tubule. There is also contribution of the cystic fibrosis transmembrane conductance regulator (CFTR) found to be crucial for renal bicarbonate (HCO3-) excretion and may directly interact with the SLC26A4 (which is already described by the authors). I would also consider including the phosphate transporter, type-II sodium-phosphate cotransporter (slc34a2 family).

Please consider the following articles:

Berg, P., Svendsen, S.L., Sorensen, M.V., Schreiber, R., Kunzelmann, K. and Leipziger, J., 2021. The molecular mechanism of CFTR‐and secretin‐dependent renal bicarbonate excretion. The Journal of Physiology, 599(12), pp.3003-3011.

Wang, J., Wang, W., Wang, H. and Tuo, B., 2021. Physiological and pathological functions of SLC26A6. Frontiers in Medicine, 7, p.618256.

Xie, Q., Welch, R., Mercado, A., Romero, M.F. and Mount, D.B., 2002. Molecular characterization of the murine Slc26a6 anion exchanger: functional comparison with Slc26a1. American Journal of Physiology-Renal Physiology, 283(4), pp.F826-F838.

Verri, T. and Werner, A., 2019. Type II Na+-phosphate cotransporters and phosphate balance in teleost fish. Pflügers Archiv-European Journal of Physiology, 471, pp.193-212.


2. The authors also mentioned Ca2+ and Mg2+ reabsorption (line 257-261). I recommend that the authors either remove this part or they really need to describe the transporters involved in Ca2+ and Mg2+ reabsorption. Also, at line 265-267 the authors mention salt wasting phenotypes in relation to only NKCC, ROMK and CLC-Kb, but there are several other transporters involved in salt transport (namely divalent ion transporters). Although the authors discuss the paracellular transport route they are missing the most crucial transcellular transporter routes for divalent ions which includes: CNNM2, CNNM4, SLC41A1-A3 and TRPM6-7 (magnesium) and PMCA1-4, NCX1-3 and TRPV5-6 (calcium). Please if you discuss the role of Ca2+ and Mg2+ include these transporters in the relevant nephron segments.

Please consider the following review papers for an overview of transporter and functions in vertebrates:

Nishimura, H. and Fan, Z., 2003. Regulation of water movement across vertebrate renal tubules. Comparative Biochemistry and Physiology Part A: Molecular & Integrative Physiology, 136(3), pp.479-498.s

Beyenbach, K.W., 2004. Kidneys sans glomeruli. American Journal of Physiology-Renal Physiology, 286(5), pp.F811-F827.

Moor, M.B. and Bonny, O., 2016. Ways of calcium reabsorption in the kidney. American Journal of Physiology-Renal Physiology, 310(11), pp.F1337-F1350.

Franken, G.A.C., Huynen, M.A., Martínez-Cruz, L.A., Bindels, R.J.M. and de Baaij, J.H.F., 2022. Structural and functional comparison of magnesium transporters throughout evolution. Cellular and Molecular Life Sciences, 79(8), p.418.

Takvam, M., Wood, C.M., Kryvi, H. and Nilsen, T.O., 2021. Ion transporters and osmoregulation in the kidney of teleost fishes as a function of salinity. Frontiers in Physiology, 12, p.664588.

3. From 296-304 you discuss NH3 and NH4 transport. Although, there are still some controversies on its regulation in the kidney I would recommend the authors to discuss the involvement of Rhesus (Rh) glycoproteins. Two types of Rh glycoproteins named Rhcg and Rhbg have been located both basolateral and apical in the distal tubule/collecting duct and implicated facilitate NH3+ and NH4+ (or both) movement. I would recommend the authors (if including ammonia regulation in the kidney) to discuss the implication of these transporters.

Please consider the following papers:

Caner, T., Abdulnour-Nakhoul, S., Brown, K., Islam, M.T., Hamm, L.L. and Nakhoul, N.L., 2015. Mechanisms of ammonia and ammonium transport by rhesus-associated glycoproteins. American Journal of Physiology-Cell Physiology, 309(11), pp.C747-C758.

Weiner, I.D. and Verlander, J.W., 2014. Ammonia transport in the kidney by Rhesus glycoproteins. American Journal of Physiology-Renal Physiology, 306(10), pp.F1107-F1120.

Weiner, I.D. and Verlander, J.W., 2017. Ammonia transporters and their role in acid-base balance. Physiological reviews, 97(2), pp.465-494.

Takvam, M., Wood, C.M., Kryvi, H. and Nilsen, T.O., 2023. Role of the Kidneys in Acid-Base Regulation and Ammonia Excretion in Freshwater and Seawater Fish: Implications for Nephrocalcinosis. Frontiers in Physiology, 14, p.1226068.

4. Many of the described transporters in the review have been associated with kidney disease. Why the focus on COVID-19? It is not that I oppose to include COVID-19 but then the authors also need to include other relevant lung conditions such chronic obstructive pulmonary disease and asthma. More important for this review would be kidney diseases which have been directly linked to dysfunction and mutations of transporters included in this review. The reader of this review would really benefit if the authors dedicated a section in which they discuss the potential implication these transporters have on kidney disease. I would recommend extending the section discussing clinical manifestation directly or indirectly related to kidney disease (from line 368 and onwards) to include transporters (mentioned in the review but also the once I have suggested above) associated with kidney disease. It could be beneficial to have a section at the end discussing the most important transporters and their association to kidney disease. Mutations and dysregulation of the following transporters have been linked to kidney stone formation, nephrocalcinosis, nephrolithiasis, defective bicarbonate reabsorption in proximal tubules, distal renal tubular acidosis, urinary bicarbonate wasting, hyperoxaluria and hypercalciuria. This includes the SLC26 family (member A1, A6, A9, A11), CFTR, SLC4A4 (or NBCe1), CAII, proton pumps (H+-ATPase; ATP6V1B1 and ATP6V0), AE1 (Cl-/HCO3- exchanger) and AQP2 to name a few.

The authors already explain the implication of mutations in transport-encoding genes which is excellent. Again, I would recommend moving the parts describing mutations, disease etc. to a dedicated chapter at the end of the review. In the different functional units/segments of the nephron I think it would be more natural to describe all the different mechanisms without dealing to much with disease. Then at the end discuss the implication of dysregulation or mutations of the described transporters and/or transport-encoding genes which will nicely summarize the physiological relevance of these at the end of the review. I would recommend the authors to also include some other potential pH sensing receptors: novel subfamilies of G protein-coupled receptors (OGR1, GPR4 and TDAG8) which have been linked to pH sensing (especially OGR1 and GPR4) and are both expressed in the kidney. The authors also mention adenylate cyclase cascade which have been suggested as a HCO3- and/or CO2 sensors which may regulate fluid and bicarbonate fluxes. If the authors decide to also include Ca2+ and Mg2+ transporters, these have also been linked to development of kidney stones, hyper/hypo-calcemia and/or -magnesia.

For all aspects described in this section please consider the following articles:

Wang, J., Wang, W., Wang, H. and Tuo, B., 2021. Physiological and pathological functions of SLC26A6. Frontiers in Medicine, 7, p.618256.

Kunzelmann, K., Ousingsawat, J., Kraus, A., Park, J.H., Marquardt, T., Schreiber, R. and Buchholz, B., 2023. Pathogenic Relationships in Cystic Fibrosis and Renal Diseases: CFTR, SLC26A9 and Anoctamins. International Journal of Molecular Sciences, 24(17), p.13278.

Rossetti, T., Jackvony, S., Buck, J. and Levin, L.R., 2021. Bicarbonate, carbon dioxide and pH sensing via mammalian bicarbonate-regulated soluble adenylyl cyclase. Interface Focus, 11(2), p.20200034.

Imenez Silva, P.H. and Mohebbi, N., 2022. Kidney metabolism and acid–base control: back to the basics. Pflügers Archiv-European Journal of Physiology, 474(8), pp.919-934.

Wagner, C.A., Kovacikova, J., Stehberger, P.A., Winter, C., Benabbas, C. and Mohebbi, N., 2006. Renal acid-base transport: old and new players. Nephron Physiology, 103(1), pp.p1-p6.


Finally here are some general comments on the structure. This is only recommendations from my side, but I think it would make the review even easier to follow. For each functional unit/segment you are describing could the authors have subtitles for each nephron segment? I think it could benefit the reader. Example of a subtitle: Proximal convoluted tubule (PCT) plays an important role in reabsorption and recycling of various ions and solutes or just the name of the segment that the authors are describing in that section. In the current structure you provide the reader with informative titles at the start (introduction, survey methodology and mechanistic topography of acid-alkaline exchange in the kidney) but once the authors describe the different functional units there is no subtitles which I personally think could be beneficial. At the end I would also recommend a title called Clinical manifestations and disorders of renal acid-base transport (or something similar). As you will see in my comments there are some important transporters missing in this review and more recent review and research papers in the field have not been included by the authors. First you may describe different clinical manifestations, but then I would discuss the consequences of dysregulation and/or mutations of the renal acid-base transporters which have been thoroughly described in the review. Although clinical manifestations in more general terms are interesting, I think it would make much more sense to focus on disorders that are connected to the specific transporters you have reviewed.

Validity of the findings

Please see my comments on the basic reporting and study design. These points needs to be addressed for this review to meet the required standards. Furthermore, I personally miss a chapter in which the authors clearly state the knowledge gaps and future direction based on the knowledge provided in the other sections. I recommend the authors to write a final section to summarise and give a future perspective on the research field. The very short conclusion as it stand currently do not address unresolved questions, knowledge gaps or future direction.

Reviewer 3 ·

Basic reporting

In this manuscript titled "Mechanisms and physiological relevance of acid-base exchange in functional units of the kidney", Elena Gantsova et al. gives a comprehensive literature review focusing on the role of the kidneys in maintaining acid-base homeostasis in the body. They discussed the importance of pH regulation and the complex mechanisms by which the kidneys contribute to this process through various physiological ion exchange processes, highlighting the roles of different parts of the kidney.

Experimental design

Major points:
1. The review appears to be thorough, encompassing a wide range of topics related to kidney function and acid-base homeostasis. However, it could benefit from a more in-depth analysis of emerging concepts and recent advancements in the field.
2. The review could integrate perspectives from related disciplines, such as nephrology, could provide a more holistic view.

Validity of the findings

3. The introduction and background sections should adequately contextualize the topic within the broader field of kidney physiology and acid-base homeostasis, providing clear relevance and rationale for the review.
4. The novelty of this review should be improved.

Reviewer 4 ·

Basic reporting

The authors have conducted a thorough review of acid-base exchange within the kidney's functional units. The introduction nicely introduces the motivation of this review paper and lay out the background for the readers.

Experimental design

The authors need to improve the logical flow of the review article to make it easier for the readers to follow:

1. Some paragraphs begin with sentences that lack a smooth transition from the preceding content, disrupting the narrative flow. The authors are encouraged to fortify the logical progression between paragraphs to aid reader comprehension. For instance, the sentence on line 172 introducing the proximal convoluted tubule (PCT) appears unexpectedly and requires a better lead-in from the preceding discussion.

2. It is essential for the opening sentence of each paragraph to function effectively as a topic sentence. In several cases, this is not achieved, which may confuse readers. For example, the paragraph starting on line 124 opens with a statement about the separate regulation of acid and base balances, but the ensuing discussion exclusively addresses the components and maintenance of acid balance.

Validity of the findings

1. The illustrations effectively depict the acid-base exchange processes, but their impact is lessened by the sole use of black and white. Employing color to differentiate ions and delineate between acid and base balance pathways would greatly enhance their educational value.

2. There are instances where chemical formulas are incorrectly formatted, lacking proper subscripts and superscripts. This is observed in line 300 with NH3 and NKCC2-, among others. Consistency in chemical notation is crucial and should be addressed throughout the document.

3. The conclusion of the article is notably succinct. Expanding this section to highlight the review's significance, its contributions to the existing body of literature, and its unique perspectives on acid-base exchange in the kidney would provide a more impactful ending.

Reviewer 5 ·

Basic reporting

1. The review focused on an interesting topic the physiological relevance of acid-base exchange in kidney. I recommend modifying the introduction with a short paragraph highlighting the importance of acid-base balance maintained by kidney and the physiological consequences of acid-base imbalance in human body.
2. Authors should maintain a similar reference pattern throughout the manuscript. Please see line no. 118 and 153. I would suggest thoroughly checking the manuscript for proper referencing, language improvement and punctuation. Please make sure that you include all recently reported relevant references.

Experimental design

1. The introduction should be enriched with recent and relevant references.

Validity of the findings

1. I would highly recommend to include a paragraph on future direction in the conclusion section.
2. I would recommend to write all diseases related to kidney mediated acid-base imbalances under a separate topic sentence or heading. It is recommended to include a paragraph on possible therapies.

Additional comments

Please mention the full forms of all abbreviations.

---

## Round 0.2 · Minor Revisions

Thank you for submitting the revised version and addressing all the reviewer's comments. There are a few minor comments from a reviewer which need to be addressed.

·

Basic reporting

The reviewers have made substantial changes and revised the manuscript to incorporate reviewer suggested changes. The manuscript is much improved from its previous version and is suitable for publication at this stage.

Experimental design

The reviewers have made substantial changes and revised the manuscript to incorporate reviewer suggested changes. The manuscript is much improved from its previous version and is suitable for publication at this stage.

Validity of the findings

The reviewers have made substantial changes and revised the manuscript to incorporate reviewer suggested changes. The manuscript is much improved from its previous version and is suitable for publication at this stage.

Additional comments

The reviewers have made substantial changes and revised the manuscript to incorporate reviewer suggested changes. The manuscript is much improved from its previous version and is suitable for publication at this stage.

Reviewer 2 ·

Basic reporting

Feedback on revised version of the manuscript “Mechanisms and physiological relevance of acid-base exchange in functional units of the kidney”

I am delighted with the revisions implemented by the authors. My feedback on the revised version is minor, and I accept the manuscript in its present form. The enhancements made to the figures are especially commendable!

General comments:
Overall, this review has significantly improved since its initial version. I would suggest that the authors carefully review the conclusion section once more. As mentioned in the specific comments, there are several sentences that could be rephrased for better clarity and impact. Additionally, I recommend revisiting and refining the section titled "Collecting Tubules and Collecting Ducts." While I understand the complexity of this topic, this section appears less structured compared to the rest of the manuscript and may benefit from further clarity, especially if the authors aim to engage researchers beyond the field. Nevertheless, the revised version is markedly enhanced, and I extend my appreciation to the authors for delivering a well-written and engaging review on the subject.

Specific comments:

Line 29-31: From “The review will be of interest” ….to “evaluate exciting studies”. The sentence is unclear and hard to read. Please insert “which” or “and” after “molecular biology, (insert) builds a strong” for clarity.

Line 58: change “constantly” to “constant”

Line 100: omit “dedicatedly”

Line 129: I’m guessing that “liquor” is supposed to be “liquid”?

Line 139-140: From “The main inorganic acids…”. I find this sentence a bit hard to read could you try to rephrase it?

Line 141: change “can be only” to “ can only be…”

Line 193 (figure 1): please omit “coordinatedly” and change “regulated” to “regulation”

Line 215: change “with” to “while”

Line 318-325: Please change the line spacing here as it is different than the rest of the manuscript

Line 326-328: Please rephrase sentence. There is a use of “extensive/extensively” and “regulating/regulated” two times in the same sentence.

Line 450: please use space to separate “forproton” to “for proton”

Line 519-520: please rephrase the sentence from “Whereas…..”

Line 616-617: Please rephrase the sentence from “In this respect….”

Line 620-621: Please repharse the sentence from “Alkali therapy may….”

Experimental design

No comment

Validity of the findings

No comment

Reviewer 3 ·

Basic reporting

I have no more questions.

Experimental design

I have no more questions.

Validity of the findings

I have no more questions.

Reviewer 4 ·

Basic reporting

The authors addressed all my comments.

Experimental design

The authors addressed all my comments.

Validity of the findings

The authors addressed all my comments.

Reviewer 5 ·

Basic reporting

The authors have implemented all the recommendations related to basic reporting.

Experimental design

The authors have implemented all the recommendations related to study design.

Validity of the findings

All the results are presented in logical and compact manner.

Additional comments

No comments.

---

## Round 0.3 · accepted · Accept

Thank you for responding to all the reviewer's comments and resubmitting the revised manuscript.